# Constitutive Heterochromatin in Eukaryotic Genomes: A Mine of Transposable Elements

**DOI:** 10.3390/cells11050761

**Published:** 2022-02-22

**Authors:** René Massimiliano Marsano, Patrizio Dimitri

**Affiliations:** 1Dipartimento di Biologia, Università di Bari, 70125 Bari, Italy; 2Dipartimento di Biologia e Biotecnologie “Charles Darwin”, Sapienza Università di Roma, 00185 Roma, Italy

**Keywords:** transposable elements, heterochromatin, *Drosophila*

## Abstract

Transposable elements (TEs) are abundant components of constitutive heterochromatin of the most diverse evolutionarily distant organisms. TEs enrichment in constitutive heterochromatin was originally described in the model organism *Drosophila melanogaster*, but it is now considered as a general feature of this peculiar portion of the genomes. The phenomenon of TE enrichment in constitutive heterochromatin has been proposed to be the consequence of a progressive accumulation of transposable elements caused by both reduced recombination and lack of functional genes in constitutive heterochromatin. However, this view does not take into account classical genetics studies and most recent evidence derived by genomic analyses of heterochromatin in *Drosophila* and other species. In particular, the lack of functional genes does not seem to be any more a general feature of heterochromatin. Sequencing and annotation of *Drosophila melanogaster* constitutive heterochromatin have shown that this peculiar genomic compartment contains hundreds of transcriptionally active genes, generally larger in size than that of euchromatic ones. Together, these genes occupy a significant fraction of the genomic territory of heterochromatin. Moreover, transposable elements have been suggested to drive the formation of heterochromatin by recruiting HP1 and repressive chromatin marks. In addition, there are several pieces of evidence that transposable elements accumulation in the heterochromatin might be important for centromere and telomere structure. Thus, there may be more complexity to the relationship between transposable elements and constitutive heterochromatin, in that different forces could drive the dynamic of this phenomenon. Among those forces, preferential transposition may be an important factor. In this article, we present an overview of experimental findings showing cases of transposon enrichment into the heterochromatin and their positive evolutionary interactions with an impact to host genomes.

## 1. Eukaryotic Transposable Elements: A Great Resource of Genome Evolution

Transposable elements (TEs) represent a significant fraction of the eukaryotic genomes, and this fraction is often referred to as the “mobilome” [1]. The mobilome comprises autonomous and non-autonomous TEs, as well as sequences derived from ancestral mobile sequences. Initially regarded as “junk DNA” [2], their role in evolution [3], genome stability [4] and structure [5], and gene regulation [6,7] is now fully acknowledged. Furthermore, TEs have become powerful tools for transgene integration and mutagenesis [8,9]. They are also progressively acquiring credit as gene therapy vectors [10,11] and as sources of ectopic gene expression tools [12].

Eukaryotic TEs are classified on the basis of their structure and transposition mechanism into two main classes (Figure 1). Class I contains retrotransposons that move via RNA transposition intermediates, whereas Class II elements transpose either using a “cut and paste” strategy (Subclass I), via rolling-circle replication [13] or single strand excision followed by extrachromosomal replication [14]. Both classes are further divided into orders, superfamilies, and families in a very complex taxonomy frame that becomes more complex as new genomes are explored.

In this review, we resume the current literature on the role of TE insertions connected to the structure and function of the heterochromatin compartment of the eukaryotic genomes.

## 2. Accumulation of TEs in Constitutive Heterochromatin of Distantly Related Eukaryotic Genomes

Constitutive heterochromatin is an almost ubiquitous component of eukaryotic genomes (up to 45% in humans [16], 30% in *Drosophila* [17], and 50–80% of some grass genomes [18,19,20]) and it is composed primarily of a variety of repetitive sequences, mostly satellite DNAs and TEs. Constitutive heterochromatin is defined as a hyper-condensed, late-replicating [21], and C-banding positive [22] genomic region, usually consisting of highly repetitive DNA. Such material has usually low gene density and is generally associated with the telomeric and pericentric regions of chromosomes. The low rate of meiotic recombination and a peculiar epigenetic status [23] are also hallmarks of the constitutive heterochromatin.

The pervasive presence and accumulation of TEs is a hallmark of heterochromatin. A build-up of TEs in this genomic region has been documented in a large number of evolutionarily distant organisms (see Table 1 for some examples). Early studies in *Drosophila* suggested evidence of a skewed distribution of TE insertions in the heterochromatin compared to euchromatin [24,25,26], and this was later confirmed in other organisms [27,28].

An extreme and intriguing example of genomic entities accumulating TEs are B chromosomes [29,30]. B chromosomes are dispensable, supernumerary, and almost heterochromatic chromosomes that are estimated to occur in approximately 15% of the taxonomic species [31]. To date, 2704 species (including 2061 plants, 14 fungi, and 629 animals; source http://www.bchrom.csic.es; last accessed on 20 December 2021) are known to contain B chromosomes that are usually riddled with TEs and display a heterochromatic state [32].

**Table 1 cells-11-00761-t001:** The accumulation of TEs in constitutive heterochromatin and their involvement in genome evolution.

**Accumulation of TEs in the Constitutive Hetrochromatin**				
**Organism**	**TE Type**	**TE Family**	**Heterochromatin Link**	**References**
*Drosophila melanogaster*	LINE	*I-element*	Preferential target	[26,33]
*Caenorhabditis elegans*	TIR (DNA)	*Tc1*	Preferential target	[34]
*Zea mais*	TIR (DNA)	*Mutator-MuDR*	Heterochromatin assembly	[35]
*Arabidopsis thaliana*	LTR	*Athila*	Preferential target	[36]
*Saccharomyces cerevisiae*	LTR	*Ty5*	Telomeric insertion	[37]
*Dictyostelium discoideum*	LTR	*Skipper-1*	Centromere insertion	[38]
*Zea mais*	LTR	*CR*	Centromere insertion	[39]
*Triticum* spp.	LTR	*CRW, Quinta*	Centromere insertion	[40]
*Tetrahymena thermophila*	TIR (DNA)	*PB*	Preferential target	[41]
*Drosophila melanogaster*	TIR	*1360*	Preferential target	[42]
*Mouse ESC*	TIR	*SB*	Preferential target	[43]
*Tetraodon nigroviridis*	TIR	*Tol2, Buffy1*	Preferential target	[44]
*Tetraodon nigroviridis*	Non-LTR	*Rex3, Babar*	Preferential target	[44]
*Drosophila melanogaster*	LTR LINE TIR	*Copia Gypsy, Mdg-1* *Blood, Doc, F, G* *Bari1, hobo*	Heterochromatic clusters Heterochromatic clusters Heterochromatic clusters	[26]
**Role of TEs in the Centromere Function**				
**Organism**	**TE Type**	**TE Family**	**Centromeric Function**	**References**
*Drosophila melanogaster*	retroelements	Several families	CENP-A recruitment	[45]
*Dictyostelium dyscoideum*	retroelements	*DIRS*	H3K9me3 and CENH3 histone marks	[38,46]
*Homo sapiens*	retroelements	Several families	Centromeric DNA organization	[47]
*basidiomycete Cryptococcus genus*	LTR	*Tcn1–Tcn6*	Centromeric DNA organization	[48]
**Birth of Satellite DNA**				
**Organism**	**TE Type**	**TE Family**	**Satellite**	**References**
*Aegilops speltoides*	LTR	*Ty3/gypsy*-like	301 bp repeat	[49]
*Arabidopsis thaliana*	CACTA	*Atenspm*	Centromeric repeats	[50]
*Arabidopsis lyrata*	LTR	*ATCOPIA93*	Centromeric repeats	[51]
*Drosophila virilis*	Not classified	*pDv*	pvB370 BamHI satellite	[52]
*Cetaceans*	Non-LTR	*L1*	Common satellite	[53]
*Oryza sativa*	LTR, DNA	Several families	Diversification of Cen8	[54]
*Drosophila melanogaster*	TIR	*Bari1*	Bari1 satellite	[55,56,57]
**Exaptation of TEs in Constitutive Heterochromatin**				
**Organism**	**TE Type**	**TE Family**	**Evolved Function**	**References**
*Drosophila melanogaster*	Non-LTR	*Het-A, TART, TAHRE*	Telomere elongation	[58,59,60,61]
*Rotifers*	retroelements	*Athena*	Telomeric repeats	[62]
Many living organisms	TIR	*pogo*	CENP-B	[63]
*Ciliates*	TIR	*PiggyBac*	PiggyMac, Pgm-like proteins—macronucleus assembly	[64,65,66]
*Arabidopsis thaliana*	LTR, DNA	*Gypsy-like, MULE*	Evolved MAIL1 and MAIN genes involved in heterochromatin assembly	[67]

### Dynamics of TE Acculumation in Constitutive Heterochromatin

The accumulation of TEs in constitutive heterochromatin of *Drosophila melanogaster* has been previously discussed in detail [26,68,69,70,71,72,73,74].

The dispersed nature of TEs in the euchromatin can be the cause of chromosomal aberrations generated through ectopic recombination between homologous TE copies [75] or by transposition-induced chromosomal breakages [76]. In addition, TEs often cause gene disruption as a consequence of their insertion or remobilization [77]. Thus, TE insertions in euchromatin are expected to be negatively selected and generally thought to be less tolerated compared to those in heterochromatin. In fact, deleterious effects of TEs in constitutive heterochromatin would be lower, or even absent, thanks to the lack of both meiotic recombination and functional genes, thus triggering a progressive accumulation of TEs into this region [78,79]. Consistent with this hypothesis, meiotic recombination negatively correlates with TE density in *D. melanogaster* [68,74], as well as in other taxa [80,81]. However, the lack of functional genes is no more considered a general feature of constitutive heterochromatin of *Drosophila* and other species. Indeed, sequencing and annotation have shown that this peculiar genomic compartment contains hundreds of transcriptionally active genes [82,83], which together account for a large fraction of the genomic territory of *D. melanogaster* constitutive heterochromatin [83].

An alternative hypothesis explaining TE accumulation in constitutive heterochromatin relies on the presence of hotspots for TE insertions. In the I-R hybrid dysgenesis, where the LINE-like I factor undergoes retrotransposition [84,85,86], heterochromatic genes were indeed found to be highly mutable targets [33], suggesting that TE accumulation may be the consequence of the preferential transposition into pericentric heterochromatin [33,71]. Accordingly, several observations suggest that the preferential TEs targeting can impact the heterochromatin establishment and maintenance. As suggested in previous reports [34,35,87], the presence of inverted repeated termini and the tendency to jump locally predispose Class II TEs to elicit small RNA-mediated silencing mechanisms that favor the formation of heterochromatic islands. In this view, large heterochromatic blocks could have arisen over evolutionary time, starting from a local accumulation of TEs. This phenomenon could also have led to the conversion of pre-existing euchromatic blocks in some species into heterochromatic sites in related species [88]. A biased transposition into “preferred” target sites can somehow justify the observed enrichment of TEs in heterochromatin. Such a preference can be reflected at the TE taxonomic level, resulting in an enrichment of certain types of TEs in the heterochromatin. Alternatively, it could be the result of the massive presence of certain types of TEs in a given genome, with other families being under-represented. As an example, LTR retrotransposons are the most abundant type of TEs in plants [89], whereas LINEs are massively present in mammalian genomes [90], especially in heterochromatin [91]. Notably, retrotransposons act as origin of replication in the heterochromatin of plant genomes, and they are typically associated with high methylation at all cytosine contexts, H3K9me2, and H3K27me1 histone marks associated with compact chromatin state [92].

At least some TEs belonging to the Chromoviridae, a lineage of the Metaviridae family [93], express a chromodomain-containing integrase that allows for interaction with nucleic acids and methylated histones [94] typical of heterochromatin, resulting in preferential insertion [95].

While the integration bias of many transposable elements has been documented, the preference for heterochromatin is less understood. Studies in plants revealed that while members of the Pseudoviridae family insert randomly, elements of the Metaviridae family preferentially target heterochromatin [36], confirming that this bias is a common feature in the Metaviridae family. *Ty5* retrotransposon of *S. cerevisiae* integrates at the telomeres and the mating type loci, two transcriptionally silent genomic regions [37]. The interaction of the *Ty5* integrase with the Sir4 protein led to a biased integration of the retrotransposon [96,97].

*Skipper-1*, a LTR-retrotransposon of *D. discoideum*, co-localizes with the centromeric histone variant cenH3 [38], although it is currently unknown as to whether the centromeric targeting of *Skipper-1* is an active CHD-mediated process or an indirect effect resulting from the loss of cells that integrate *Skipper-1* in other gene-rich genomic regions.

In plants, a specific family of *Ty3/gypsy* retrotransposons, the CR family, preferentially inserts into the centromeric satellite DNA and associates with the centromere-specific protein CenH3, suggesting an active role in the centromere function and kinetochore formation [39,40].

However, many other transposon types display a biased insertion pattern toward heterochromatin.

The domesticated transposase Tpb2pn evolved from the PB transposase [98] associates with heterochromatin bodies [41], in which the internal eliminated sequences (IES) are assembled and subsequently removed to form the genome of the vegetative macronucleus of the ciliated protist *Tetrahymena thermophila* [99].

In some cases, the introduction of TE-related sequences in the integration cassette changes the integration pattern of the transgenes, as demonstrated for the *1360* element in *D. melanogaster* [42], suggesting a homing-like behavior of TE-related sequences. 

Even the SB transposase, resurrected from a fish Tc1-like transposon [100], preferentially localizes to ectopic heterochromatin sites induced with a tetracycline-controlled trans-repressor protein in mouse embryonic stem cells [43].

Preferential accumulation in the heterochromatin of several TE families has been also observed in *Tetraodon nigroviridis* [44,101]. It has been speculated that heterochromatin could represent a reservoir of TEs [44] in the pufferfish genome. This hypothesis finds support in studies performed in plants, demonstrating that in the developing gametophytes the reprogramming of heterochromatin leads to the reactivation of heterochromatic TEs [102].

## 3. Positive Interactions between Transposable Elements and Constitutive Heterochromatin in Different Host Genomes

### Transposable Elements and Transcription of Heterochromatic Genes

As discussed in the previous section, the picture of constitutive heterochromatin as the silent part of the genomes should be profoundly reconsidered, at least in *D. melanogaster* [83,103]. Indeed, the role of transcripts stemming from heterochromatin is well-recognized in many organisms [103,104]. 

In *D. melanogaster*, TE copies are intimately associated with the heterochromatic genes’ body, both in the flanking and intronic regions [82,105,106]. The presence of large introns packed with structurally degenerate TEs represents a hallmark of *D. melanogaster* heterochromatic genes [107,108], especially those found in the Y chromosome, which are essential to male fertility [109,110].

Thus, it may be possible that during evolution, TE sequences became functionally integrated within the genes in heterochromatin, acting as regulatory elements that drive gene expression by recruiting specific epigenetic factors, such as HP1 protein. 

Indeed, constitutive heterochromatin of *D. melanogaster* constitutes a relevant case study of epigenomic conflict. While heterochromatin contains genes that are expressed throughout the development and across tissues, it is tagged with repressive epigenetic marks, such as methylation of H3K9 and H4K20 [111], in addition to the transcriptional permissive histone modifications [83]. 

However, transcription of heterochromatic sequences is not limited to *Drosophila* species. Studies performed in the last decades are consolidating a new perspective of constitutive heterochromatin on the basis of its transcriptional plasticity. Heterochromatin transcription indeed plays a critical role in establishing heterochromatin de novo in the daughter cell after mitosis completion [104], as well as during early embryo development in mammals [112]. Although specific studies are currently lacking, it is conceivable that heterochromatic TE copies can also play a significant role in the transcriptional regulation of heterochromatin. 

## 4. TE Exaptation in a Heterochromatin-Related Context

Exaptation [113] is an evolutionary phenomenon that co-opts genetic entities to new functions that aid the host genome’s performance. This shifting in traits’ function frequently involves TEs [114], and in particular, several functions related to heterochromatin have evolved from TEs.

### 4.1. Transposable Elements and Telomeres

Very special examples of TE domestication associated with neofunctionalization are the telomere maintenance in *D. melanogaster*, which has been extensively discussed [59,60,61]. The elongation mechanism relies entirely on the selective transposition of three L1-like TE families, *Het-A*, *TART*, and *TAHRE* [58], that avoid chromosome consumption at their ends (Table 1). 

While this example seems to be limited to the species of the *Drosophila* genus, this recalls a theory on the telomere origin that suggests that group II introns (a class of bacterial mobile elements) could have originated the ancestral eukaryotic telomeres, allowing the formation of primitive t-loops [115], suggesting a TE-based origin of the telomeres.

TE enrichment in telomeric and sub-telomeric regions has been described in diverse species of fungi [116], vertebrates [117], insects, protozoa [118], and plants [119]. However, differently from the *Drosophila* telomeric TEs, there is no reported function for the presence of TEs in the telomeres of other species, suggesting an accumulation resulting from the absence of selective pressure at these loci.

Notably, members of the Athena clade of the Penelope-like retrotransposons identified in Rotifer (Bdelloidea) lack the endonuclease domain, contain short stretches of telomeric repeats at their 3′ end, and are preferentially oriented toward the telomere with their 5′ truncated end [62].

### 4.2. Transposable Elements and Centromeres

The centromere is the major locus buried in the constitutive heterochromatin. The hallmark of the centromeric DNA, virtually in all eukaryotic chromosomes, is the enrichment in satellite DNA, but very often centromeres are associated with a high frequency of TE insertions that built up the architecture of such complex chromosomal structures [120] (Table 1).

One of the well-known cases of TE exaptation connected to the centromere function is the CENP-B protein [63]. CENP-B is a widely conserved centromere-binding protein formerly found in mammals that also has homologues in non-mammalian species [121], including yeast [122]. The CENP-B protein localizes densely at the centromere of all human chromosomes but the Y chromosome, and it is involved in chromosome segregation [123] and also required for kinetochore nucleation [124]. It has been proposed that the CENP-B protein evolved from an ancestral *pogo*-like transposase [125,126] and that its recruitment occurred at least twice during evolution [121,127]. The evolutionary history of CENP-B is acknowledged as one of the most interesting cases of convergent TE domestication [121].

Whether TEs play a pivotal role in either establishing a functional centromere, evolving new centromeres from scratch, or generating new satellite DNAs are still unsolved questions.

The massive presence of TEs in the centromeric DNA of *D. melanogaster* was first highlighted using combined sequencing and chromosomal deletions analyses [128]. This study mapped the smallest DNA sequences sufficient for centromere function to a 420 kb region containing the AAGAG and AATAT satellites interspersed with “islands” of complex sequences, such as TEs. Further studies confirmed that *Drosophila* centromeres range between approximately 200 and 500 kb in size [129] and are highly enriched in tandem repeats [128,130]. Recently, Chang et al. [45], by mapping CENP-A on single chromatin fibers at high resolution, reported that the CENP-A primarily associates with islands of retroelements that are flanked by satellite DNA. In addition, they demonstrated that the *G2*/*Jockey-3* retroelement is the most highly enriched sequence in CENP-A chromatin, and it is shared among all centromeres. Since this feature is somehow conserved in related species with divergent centromeric satellites, these results strongly suggest a conserved role of retroelements in centromere specification and function in *Drosophila*.

The massive occupancy of TEs in the centromeric and pericentromeric DNA regions of the chromosome is also a shared feature in other species. An 86% fraction of the 170–360 Kbp long *Dictyostelium dyscoideum* centromeric DNA is composed of LTR retrotransposons [46]. Nearly half of the centromeric sequences are represented by the DIRS element, which seems to be a centromere-specific element co-localizing with CENH3 and H3K9me3 [38].

Human centromeric and pericentromeric regions also appear to constitute “soft-landing” platforms for TEs insertions. Indeed, a recent investigation of TE insertions in 5675 genomes has revealed that the preferred insertion sites of LINE elements lay within centromeric DNA [131]. However, in some cases TEs insertions appear to be excluded from centromeric DNA regions in humans. The recent development of the long reads sequencing methods [132,133,134] allowed for the determination of the human chromosome 8 centromeric and pericentromeric DNA sequence [47]. The centromeric DNA of the human chromosome 8 consists of five major evolutionary layers, showing a peculiar mirror symmetry. Each layer consists of sequences showing progressively higher sequence similarity from the outermost to the innermost. With respect of this organization, TE insertions can be only found in the outermost layer, wherein they are interspersed with monomeric and divergent α-satellite [47]. While the same organization of the centromeric and pericentromeric regions has not been observed for other chromosomes (e.g., chromosome X), it could still suggest a functional role.

A possible explanation of the target preference and accumulation of TEs in the centromeric DNA comes from studies in fungi. In three species of the pathogenic basidiomycete *Cryptococcus* genus, the presence of full-length TEs is observed only in species with long centromeres and in which the RNAi process is active (i.e., *C. neoformans*, and *C. deneoformans*). By contrast, *C. deutereogatti*, which lacks the RNAi pathway, has short centromeres void of active TEs [48]. Since a similar relationship between the presence of RNAi and centromere length exists in species of other pathogenic basidiomycetes [48] it has been suggested that the loss of RNAi could increase recombination between transposons and promote loss of full-length elements.

The amplification of pre-existing TE copies in the constitutive heterochromatin may also have contributed to the birth of pericentromeric repeats.

In the plant *Aegilops speltoides*, the 250 bp centromeric satellite shares high similarity to portions of a *Ty3/gypsy*-like retrotransposons [49]. TEs homologous to centromeric repeats have also been identified in other plant species, such as the Atenspm element in *A. thaliana* [50] and ATCOPIA93 in *A. lyrata* [51], as well as in animals such as the pDv element in *D. virilis* [52], and in cetaceans [53].

The compact organization of the centromere 8 in rice variety Nipponbare, which contains 65 Kbp of repeats [135], and the availability of sequencing data from multiple rice varieties [136] offered the opportunity to investigate centromeric transposons and their role in centromere evolution [54]. These studies revealed the role of TEs in both the structure and the rapid diversification of the Cen8 sequences between the cultivated rice species.

The ability of TEs to form tandem repeats is well documented in *Drosophila*, with at least two well-studied examples involving Class II TEs. The first example concerns tandem repeats of the *P-element*, which are frequently generated during genetic screens in *D. melanogaster* [137] and naturally found in *D. guanche* [138]. It has been demonstrated that *P-element* tandem repeats are formed by double insertion at the same site [57]. The second example concerns the *Bari1* element, which is arranged as a regular tandem repeat in the heterochromatin [139]. Rolling circle replication, after circularization of an excised element, has been proposed as a possible mechanism to explain this arrangement [56]. This mechanism explains the origin of the two evolutionarily fixed *Bari1* clusters in the deep heterochromatin of *D. melanogaster*, mapping on the second and the X chromosomes correspondingly [55,57].

In this context, it has been suggested that satellite DNA can be generated from transposons following different routes. One is the accumulation of TEs that can be detected as long stretches of DNA sequences similar to TEs. The observed sequence similarity between some TEs and satellite DNA in several species is indeed one of the main observations that has led to the intriguing hypothesis that satellite DNA may have derived from TEs. This suggestion is supported by observations in *Drosophila* [52] and humans [140]. Alternatively, satellite DNA can be formed from internal repeats residing into the TE itself. Tandem repetitions can be found in all types of TEs [141], which can virtually generate satellite DNA.

### 4.3. Additional Examples of TE Exaptation in Constitutive Heterochromatin

In the ciliate *Paramecium* species, the domesticated PiggyBac transposase PiggyMac (Pgm) protein [64,65] and the Pgm-like proteins [66] are involved in the programmed elimination of thousands DNA sequences that are at the basis of the macronucleus origin.

In *Arabidopsis*, genetic loss of two genes MAIL1 and MAIN result in impaired condensation of pericentromeric heterochromatin and upregulation of TE transcription, suggesting a transcriptional repression role. The proteins encoded by MAIL1 and MAIN appear to be derived from a subset of *Ty3/gypsy* retrotransposons found in angiosperms [67].

An additional role of TEs in the heterochromatin appears to be related to the embryo development. In a recent study, the upstream region of retrotransposons belonging to few active families was identified as the nucleation sites of heterochromatin formation during early stages of embryo development in the fly [142,143], and similar observations have been made in mammals [144].

## 5. Potential Implications of TE Heterochromatic Copies in Diseases and Aging

Being habitual residents of heterochromatin, TEs are obviously linked to human diseases related to heterochromatin defects, since loosening in heterochromatin compaction can cause the de-repression of TEs, thus leading to deleterious physiological changes such as aging, cancer, and neurological disorders. Although it is difficult to establish the direct link between heterochromatic TE copies and the disease itself, some recent papers suggest that this could be the case.

It is well established that the conformation of the chromatin in the interphase nucleus has a physiological relevance to cell life. In this view, any perturbation of the chromatin organization in the 3D nucleus can contribute to cell response to environmental stress or to the onset of diseases. 

Laminopathies are heterogeneous diseases caused by dysfunction of the LAMIN proteins that ensure the correct anchoring of heterochromatin to the nuclear periphery [145].

Heterochromatin relaxation and TE de-repression have been related to the onset of ALS [146] and tauopathies [147]. Although the role of TE is not clear in the latter, HERVKs appear to be activated by heterochromatin loosening, which causes neuronal death [148].

In a *Drosophila* tauopathy model, Tau dysregulates TEs resident in the heterochromatin, reinforcing the above-described observation in mammals [148].

Additionally, chromatin undergoes global chromatin compaction upon activation of cell motility in several cell types of cancer metastasis [149].

Similarly, an aging-dependent loss of heterochromatin induces a dysregulation of TEs that can in turn produce genome instability and activation of inflammatory responses [150].

Although more experimental evidence is needed, there is a clear connection between the role of heterochromatic copies of TEs and the onset of pathological phenotypes.

## 6. Conclusions

About 80 years have passed since Barbara McClintock published her results on the controlling elements in maize, which were considered very controversial at that time by the scientific community. 

Today, in the post-genomic era, due to the huge amount of genome sequences, transposable elements are still arousing a large amount of interest in the field of genome evolution. In particular, transposable elements are no longer considered only a mere example of genomic parasites, but it is widely recognized that they have colonized all eukaryotic genomes and represent a major force driving the evolution of organisms. 

The development of new genome sequencing methods and annotation protocols could enforce this vision, opening the possibility of comparing constitutive heterochromatin in thousands of eukaryotic organisms. Such studies are needed to shed light on the dynamics of TE accumulation in the heterochromatin and to resolve the role of TEs in the determination of one of the most elusive chromosome structures, i.e., the centromere. Large-scale comparative analysis will also help determine the impact of TEs on heterochromatic gene expression and to disentangle the complex evolutionary trajectories that lead to TE exaptation and domestication. Since many of these issues are (or they will be) related to human health/diseases, the role of TEs in the constitutive heterochromatin is likely to be even more substantial than we may now imagine, and a multiplicity of their roles and impact on cellular functions and genome evolution will come to light.

## Figures and Tables

**Figure 1 cells-11-00761-f001:**
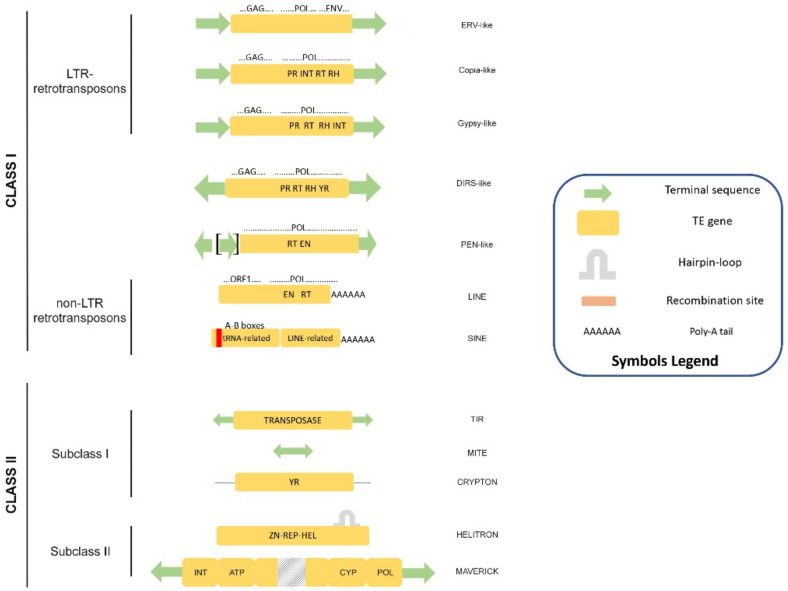
Classification of eukaryotic transposable elements according to Wicker et al. [15]. The structure and the coding potential are depicted for each of the superfamilies. Symbols are explained in the legend box.

## Data Availability

Not applicable.

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
