# Peer review of "Constitutive Heterochromatin in Eukaryotic Genomes: A Mine of Transposable Elements"

_cells, 2022, doi:10.3390/cells11050761_

Round 1

Reviewer 1 Report

The manuscript entitled “Constitutive heterochromatin in eukaryotic genomes: a mine of transposable elements” by René Massimiliano Marsano and Patrizio Dimitri, addresses an interesting subject, and reviews the evidence that TEs play an important role in heterochromatin function and evolution from a diversity of organisms such as plants, insects and mammals. The manuscript should, however, be carefully reviewed regarding the use of English grammar. There are also a few very long sentences that make the manuscript difficult to read. Some examples are:

29 – “with an impact” not “with and impact”

55 – “for each of the superfamilies” and not “each of the superfamily”

80 – “in detail” not “in details”

97 – “undergoes” not “undergo”

104 – “blocks” not “block”

221 - delete “of”

Please take special attention to scientific names: Drosophila, Cryptococcus, Paramecium, and Arabidopsis should be italicised.

In table 1 scientific names should not be abbreviated.

Difficult to read/understand sentences:

81-82 – First explain how TEs cause chromosomal aberrations and only then how they cause gene disruption.

117-125 – Should be a single paragraph.

213-220 – Please give more details.

248-251 – Please clarify. Difficult to read.

341-346 Split sentence into several to facilitate reading.

The title of the first section (1. Introduction) is also awckward, since this is a very short section mostly on TE classification, and thus I believe it should be called “TE classification”

I also feel that the “Conclusion” should have a stronger “Take home message”, that should summarize the main findings of each of the preceding sections.

Author Response

REPLY TO REVIEWER 1

We thank this reviewer for the helpful comments provided. Please, find below a point-by-point response to the comments and concerns raised (our response in italic).

29 – “with an impact” not “with and impact”

Changed accordingly

55 – “for each of the superfamilies” and not “each of the superfamily”

Edited

80 – “in detail” not “in details”

Corrected

97 – “undergoes” not “undergo”

Fixed

104 – “blocks” not “block”

Corrected

221 - delete “of”

Deleted

Please take special attention to scientific names: Drosophila, Cryptococcus, Paramecium, and Arabidopsis should be italicised.

We have revised all the taxonomic names throughout the manuscript

In table 1 scientific names should not be abbreviated.

Revised

Difficult to read/understand sentences:

81-82 – First explain how TEs cause chromosomal aberrations and only then how they cause gene disruption.

We have revised the text according to this suggestion

117-125 – Should be a single paragraph.

We have revised the paragraph

213-220 – Please give more details.

We have restructured the paragraph providing with more details on the CENP-B origin

248-251 – Please clarify. Difficult to read.

We have reworded this part to enhance the readability

341-346 Split sentence into several to facilitate reading.

The text was revised accordingly

The title of the first section (1. Introduction) is also awckward, since this is a very short section mostly on TE classification, and thus I believe it should be called “TE classification”

Thank you for noticing this mistake. The proper title of this paragraph was erroneously lost during the initial manuscript formatting. We have now fixed this error.

I also feel that the “Conclusion” should have a stronger “Take home message”, that should summarize the main findings of each of the preceding sections.

We have revised this paragraph trying to highlight the main issues considered in the main text.

Reviewer 2 Report

In this manuscript, the authors review the recent findings on the role of transposable elements in the evolution and functioning of heterochromatin. The review is an important contribution to the field. It is well written and supported by very informative tables and relevant references.

Below, a list of minor comments to the manuscript:

1. Line 59. Provide here the definition of “constitutive heterochromanin”.

  1. Table 1, the subsection “Exaptation of TEs in constitutive heterochromatin”. It is better to reformulate the heading as “Exaptation of TEs in heterochromatin”. Telomeric heterochromatin rather does not belong to constitutive heterochromatin; it has many features of active chromatin.
  2. Line 103. The phrase should be specified as follows “to elicit SMALL RNA-MEDIATED silencing mechanisms”.
  3. Line 116. Some short explanation is required, e.g. “and enrichment by H3K9me2 and H3K27me1 HISTONE H3 MARKS ASSOCIATED WITH COMPACT CHROMATIN STATE”.
  4. Lines 146-147. This phrase should be more detailed, e.g. “fish SB transposase reconstructed in human cells…”
  5. Line 176. “transcriptionally plasticity” should be changed to “transcriptional plasticity”
  6. Line 218-219. This phrase should be specified as follows “…the recruitment of pogo- like transposase TO BECOME CENTROMERE-BINDING PROTEIN occurred…”
  7. Line 277. Should be specified: “two well-studied examples FOR CLASS II TEs”
  8. The paragraph, lines 282-290, is confused and should be rephrased for clarity. In this shape, the difference between two scenarios of satellite DNA origination from TEs is ambiguous.
  9. Line 325. There is a typo, should be “loss of heterochromatin induces…”
  10. Check carefully the reference list. In pdf version, the text in references 39, 40 and 134 is abnormally formatted. Refs 31, 34, 61, 74, 82, 84, 91, 110, 111, 138 are truncated. A fragment of Ref 70 is highlighted by yellow.

Author Response

REPLY TO REVIEWER 2

We thank this reviewer for the helpful comments provided. Please, find below a point-by-point response to the comments and concerns raised (our response in italic).

  1. Line 59. Provide here the definition of “constitutive heterochromanin”.

We have added the definition as requested.

Table 1, the subsection “Exaptation of TEs in constitutive heterochromatin”. It is better to reformulate the heading as “Exaptation of TEs in heterochromatin”. Telomeric heterochromatin rather does not belong to constitutive heterochromatin; it has many features of active chromatin.

We respectfully disagree with the reviewer's point. There is plenty of paper remarking that constitutive heterochromatin includes the telomeric regions and that it shares similar hallmark epigenetic modifications to those present at centric heterochromatin. We therefore prefer to maintain the original header for the aforementioned table subsection.

As an example, below we quote some statements from relevant literature (concerning Drsosophila and mammals) and from a textbook.

  • "Constitutive heterochromatin is a major component of the eukaryotic nucleus and is essential for the maintenance of genome stability. Highly concentrated at pericentromeric and telomeric domains, heterochromatin is riddled with repetitive sequences and has evolved specific ways to compartmentalize, silence, and repair repeats". From: Janssen, Colmenares and  Karpen Heterochromatin: Guardian of the Genome Annu Rev Cell Dev Biol (2018)

  • "After cell division, regions forming heterochromatin remain condensed, especially around the centromeres and telomeres (called constitutive heterochromatin)"From: Introduction to Genetic Analysis, 11th edition Griffiths, Wessler, Carrol, and Doebley (2014)

  • "...telomeric chromatin shares similar hallmark epigenetic modifications to those present at pericentric heterochromatin and those modifications are maintained by the Suv39h1 and Suv39h2 HMTases". From: Blasco Carcinogenesis (2004).

Line 103. The phrase should be specified as follows “to elicit SMALL RNA-MEDIATED silencing mechanisms”.

we have modified the text as suggested

Line 116. Some short explanation is required, e.g. “and enrichment by H3K9me2 and H3K27me1 HISTONE H3 MARKS ASSOCIATED WITH COMPACT CHROMATIN STATE”.

Changed as suggested

Lines 146-147. This phrase should be more detailed, e.g. “fish SB transposase reconstructed in human cells…”

We have added details to this example

Line 176. “transcriptionally plasticity” should be changed to “transcriptional plasticity”

Corrected

Line 218-219. This phrase should be specified as follows “…the recruitment of pogo- like transposase TO BECOME CENTROMERE-BINDING PROTEIN occurred…”

We have rebuilt this sentence also according to the comments of reviewer 1

Line 277. Should be specified: “two well-studied examples FOR CLASS II TEs”

Changed accordingly

The paragraph, lines 282-290, is confused and should be rephrased for clarity. In this shape, the difference between two scenarios of satellite DNA origination from TEs is ambiguous.

We have restructured this part to clarify the two examples described

Line 325. There is a typo, should be “loss of heterochromatin induces…”

Removed an extra word

Check carefully the reference list. In pdf version, the text in references 39, 40 and 134 is abnormally formatted. Refs 31, 34, 61, 74, 82, 84, 91, 110, 111, 138 are truncated. A fragment of Ref 70 is highlighted by yellow.

All the references have been carefully revised and properly formatted

Round 2

Reviewer 1 Report

line 295 - delete the and in "and. It has"

Everything else is fine.